MxaY regulates the lanthanide-mediated methanol dehydrogenase switch in Methylomicrobium buryatense

Chu Frances 1
Beck David A.C. 1 2
Lidstrom Mary E. lidstrom@uw.edu 1 3
1 Department of Chemical Engineering, University of Washington , Seattle , WA , United States
2 eScience Institute, University of Washington , Seattle , WA , United States
3 Department of Microbiology, University of Washington , Seattle , WA , United States
Read Timothy
Electronic publication date: 2016 Sep 7
Publication date: 2016
Volume: 4
Electronic Location ID: e2435
Received 2016 May 25; Accepted 2016 Aug 12
Copyright: ©2016 Chu et al.
Copyright year: 2016
Copyright holder: Chu et al.
License: This is an open access article distributed under the terms of the Creative Commons Attribution License, which permits unrestricted use, distribution, reproduction and adaptation in any medium and for any purpose provided that it is properly attributed. For attribution, the original author(s), title, publication source (PeerJ) and either DOI or URL of the article must be cited.
License URL: https://creativecommons.org/licenses/by/4.0/

Keywords: Methanol dehydrogenase, XoxF, Lanthanides, MxaY, MxaFI

Funding: DOE ARPA-E DE-AR0000350 This work was supported by a grant from DOE ARPA-E (DE-AR0000350) to ME Lidstrom. The funders had no role in study design, data collection and analysis, decision to publish, or preparation of the manuscript.

==============================
Many methylotrophs, microorganisms that consume carbon compounds lacking carbon–carbon bonds, use two different systems to oxidize methanol for energy production and biomass accumulation. The MxaFI methanol dehydrogenase (MDH) contains calcium in its active site, while the XoxF enzyme contains a lanthanide in its active site. The genes encoding the MDH enzymes are differentially regulated by the presence of lanthanides. In this study, we found that the histidine kinase MxaY controls the lanthanide-mediated switch in Methylomicrobium buryatense 5GB1C. MxaY controls the transcription of genes encoding MxaFI and XoxF at least partially by controlling the transcript levels of the orphan response regulator MxaB. We identify a constitutively active version of MxaY, and identify the mutated residue that may be involved in lanthanide sensing. Lastly, we find evidence to suggest that tight control of active MDH production is required for wild-type growth rates.

Introduction

Interest in the use of methanotrophs, or methane-oxidizing bacteria, as biological catalysts for the production of chemicals or biofuels has increased as atmospheric levels of methane rise. Methanotrophs oxidize and utilize methane as their carbon and energy source, and the industrial utilization of methanotrophs is an attractive way to mitigate atmospheric methane (Fei et al., 2014; Haynes & Gonzalez, 2014; Kalyuzhnaya, Puri & Lidstrom, 2015). New genetic tools and the discovery of rapidly-growing methanotrophic strains has enabled rapid iteration of metabolic engineering strategies (Puri et al., 2015; Yan et al., 2016). The deployment of methanotrophs for industrial use is dependent on successful metabolic engineering strategies, which requires an in-depth knowledge of methanotrophic metabolism.

Methanotrophs are a subset of microorganisms that are capable of methylotrophy, the metabolic mode in which reduced carbon substrates containing no carbon–carbon bonds are converted into energy and biomass (Anthony, 1982; Lidstrom, 2006; Chistoserdova, 2011). In methanotrophs, reducing power is generated when methane is first oxidized to methanol and then to formaldehyde. Formaldehyde can be reduced further to formate and carbon dioxide to generate more reducing power, or it can be assimilated into biomass (Chistoserdova, 2011). Type I methanotrophs, which are gammaprotebacteria, assimilate formaldehyde into biomass using the highly efficient ribulose monophosphate (RuMP) pathway (Kemp & Quayle, 1966; Kemp & Quayle, 1967; Kalyuzhnaya et al., 2013). Type I methanotrophs are particularly suited for use as industrial microorganisms as this assimilatory pathway allows for the most efficient conversion of methane into value-added chemicals (Kalyuzhnaya, Puri & Lidstrom , 2015).

In the majority of methanotrophs identified, the first two reactions required to convert methane into biomass and energy are carried out by virtually functionally redundant enzymes, whose gene transcription is regulated by the metals contained in the enzyme active sites (Semrau, DiSpirito & Yoon, 2010; Chistoserdova, 2011; Keltjens et al., 2014; Farhan Ul Haque et al., 2015b; Chu & Lidstrom, 2016). Functional redundancy between these enzyme complexes may be crucial in fluctuating environments with uncertain metal availability. The tight transcriptional regulation of these enzyme complexes by active site metals suggests that the unnecessary production of inactive enzymes is metabolically costly to cells.

Methane oxidation in most methanotrophs can be carried out by two separate, differentially-regulated methane oxidation systems: the copper-dependent particulate methane oxidation (pMMO) system and the iron-dependent soluble methane oxidation (sMMO) system (Semrau, DiSpirito & Yoon, 2010; Chistoserdova, 2011). The presence of copper regulates the transcription of these two systems, with genes encoding pMMO being induced only in the presence of copper (Nielsen, Gerdes & Murrell, 1997; Choi et al., 2003). Under these conditions, the genes encoding the sMMO system are downregulated.

Similarly, methanol oxidation can be carried out by two differentially-regulated methanol dehydrogenase (MDH) enzymes. Both MDH enzymes are periplasmic and contain the pyrroloquinoline quinone (PQQ) cofactor. The canonical, well-studied MDH is a heterotetrameric enzyme composed of two large subunits, MxaF, and two small subunits, MxaI, with a calcium ion in the MxaF active site. The more recently-discovered XoxF MDH is reported to be composed of a homodimer of XoxF subunits (Harms et al., 1996; Chistoserdova & Lidstrom, 1997; Schmidt et al., 2010; Chistoserdova, 2011; Pol et al., 2014). Purified XoxF enzymes have been found to contain lanthanides, rather than calcium, in the active site (Pol et al., 2014).

Similar to copper regulation of genes encoding methane monooxygenases, lanthanides differentially regulate the transcription of genes encoding the MDH enzymes (Farhan Ul Haque et al., 2015b; Skovran & Martinez-Gomez, 2015; Chu & Lidstrom, 2016; Gu et al., 2016; Vu et al., 2016). The gene encoding XoxF is transcribed in the presence of lanthanides and the operon encoding the MxaFI genes is transcribed in the absence of lanthanides. Moreover, XoxF has been purified as the dominant MDH when methylotrophs are cultivated in the presence of lanthanides (Fitriyanto et al., 2011; Hibi et al., 2011; Nakagawa et al., 2012). As concentration of lanthanides in the environment is high enough to favor xoxF transcription, XoxF is likely the dominant MDH operant under environmental conditions in some methylotrophic species (Delmotte et al., 2009; Grob et al., 2015; Ramachandran & Walsh, 2015; Skovran & Martinez-Gomez, 2015; Chu & Lidstrom, 2016; Vu et al., 2016).

Numerous regulators have been found to control the transcription of genes encoding the MDH enzymes. In the non-methanotrophic methylotroph Methylobacterium extorquens AM1, the mxa operon is controlled by at least two two-component systems, MxcQE and MxbDM, and the orphan response regulator MxaB (Morris & Lidstrom, 1992; Xu et al., 1995; Springer, Morris & Lidstrom, 1997; Springer, Auman & Lidstrom, 1998). The presence of xoxF1 and xoxF2 genes are also required for MxaF protein expression in M. extorquens AM1 (Skovran et al., 2011). The differential regulation of the mxaFI and xoxF genes is at least partly carried out by the MxbDM two-component system in M. extorquens AM1 (Skovran et al., 2011) and by the orphan response regulator MxaB in M. buryatense (Chu & Lidstrom, 2016). These regulators simultaneously increase mxa expression while decreasing xoxF expression. Like mxaFI transcription, transcript levels of mxaB are also controlled by lanthanides (Chu & Lidstrom, 2016).

In the non-methanotrophic methylotroph Paracoccus denitrificans, the three-gene operon mxaXYZ located upstream and divergently transcribed from the mxaF operon, was required for growth on methanol (Harms et al., 1993). The response regulator encoded by the mxaX gene was shown to be required for mxaF gene expression (Yang et al., 1995). Although the histidine kinase encoded by mxaY was dispensable for mxaF gene expression, the expression of a plasmid-based copy of mxaY resulted in enhanced mxaF gene transcription (Yang et al., 1995).

M. buryatense strain 5GB1C is an attractive model for the study of the lanthanide-mediated MDH switch. It is a fast-growing Type I methanotroph with numerous genetic tools, making it a desirable platform for development as an industrial microorganism (Puri et al., 2015; Yan et al., 2016). Unlike many other methylotrophic organisms that harbor multiple copies of the MDH genes, the M. buryatense genome contains only one copy of genes encoding MxaFI and XoxF (Khmelenina et al., 2013). This allows us to easily manipulate and study potential factors involved in the lanthanide-mediated MDH switch.

In this study, we identified an MxaY homologue in M. buryatense, which functions to increase the transcription of mxaB and the mxaF operon while downregulating xoxF transcription in the absence of lanthanum. We identified a single nucleotide polymorphism (SNP) in MxaY which renders the histidine kinase MxaY constitutively active. Phenotypic analyses of the M. buryatense ΔmxaY mutant suggest that the histidine kinase plays pleiotropic roles in lanthanide-mediated cellular response.

Materials and Methods

Strains and growth conditions

M. buryatense 5GB1C and its derivatives were grown in a modified NMS medium with methane, as previously described (Puri et al., 2015). Supplements were added as follows: 2.5% (weight/volume) sucrose, 100 µg hygromycin/ml, 50 µg kanamycin/ml, 30 µg zeocin/ml, and 30 µM lanthanum chloride (Sigma; lanthanum in 3+ oxidation state). All culturing glassware for experiments performed without lanthanum was acid-washed overnight in 1 M hydrochloric acid before use, which we have shown is sufficient to prevent a lanthanide cellular response (Chu & Lidstrom, 2016). All strains used in this study are listed in Table 1. The M. buryatense 5G genome sequence is deposited in GenBank/EMBL under the accession numbers AOTL01000000, KB455575, and KB455576 (Khmelenina et al., 2013).

Table 1 M. buryatense 5GB1C strains and plasmids used in this study.

Strain name	Genotype	Antibiotic resistance	
M. buryatense 5GB1C	Wild-type	None	
FC53	ΔxoxF (Chu & Lidstrom, 2016)	Unmarked	
FC57	ΔxoxFS::zeoR (Chu & Lidstrom, 2016)	Zeocin	
FC70	ΔMETBUDRAFT_1817::zeoR	Zeocin	
FC74	ΔmxaY	Unmarked	
FC77	ΔmxaY; METBUDRAFT_2794::PmxaY-mxaY	Unmarked	
FC78	ΔmxaY; METBUDRAFT_2794::PmxaY-mxaY E147G	Unmarked	
FC93	ΔxoxFS ΔmxaY E147G; METBUDRAFT_2794::PmxaY-mxaY	Unmarked	
Plasmids			
pFC44	METBUDRAFT_2794::PmxaY-mxaY	Kanamycin	
pFC45	METBUDRAFT_2794::PmxaF-mxaY E147G	Kanamycin	

DNA isolation and whole genome resequencing of ΔxoxFS

The ΔxoxFS strain was isolated as previously described (Chu & Lidstrom, 2016). Genomic DNA from this strain was isolated using a phenol chloroform extraction. The genome resequencing was performed on an Illumina MiSeq with Illumina’s 300-bp paired-end (PE) protocol by MR DNA lab (Shallowater, TX, USA) with multiplexing. The paired-end reads for each condition were pooled and processed with breseq version 0.27.1 (Deatherage & Barrick, 2014) against the MaGE (Vallenet et al., 2013) MBURv2 annotation of M. buryatense 5G from 26 November 2013.

Genetic manipulations

Gene knockout constructs were composed of assembled PCR products and electroporated into M. buryatense5GB1C as previously described (Chu & Lidstrom, 2016; Yan et al., 2016). For construction of the final ΔmxaY mutant and ΔxoxFS ΔmxaY mutant, the zeocin resistance marker was excised, leaving an unmarked mxaY deletion with a single flippase (FLP) recombinase target (FRT) site remaining, using the FLP-FRT site-specific recombination system (Chu & Lidstrom, 2016).

All complementation strains were constructed by conjugation of a pCM433kanT-based suicide plasmid harboring PmxaY and a wild-type or E147G mutant copy of the mxaY open reading frame, into the appropriate knockout strain (Puri et al., 2015). Conjugation was performed using Escherichia coli S17-1 λpir as the donor strain (Puri et al., 2015). This complementation method results in the integration of the mxaY construct into a region of the M. buryatense 5GB1C chromosome known to be transcriptionally silent (between genes METBUDRAFT_2794 and METBUDRAFT_2795), allowing for native transcription of the inserted mxaY gene (Yan et al., 2016). Sucrose counter-selection was employed to unmark all complementation constructs (Puri et al., 2015). Strain FC74 was conjugated with plasmid pFC44 to create the wild-type mxaY complementation strain (FC77) or with plasmid pFC45 to create the E147G MxaY complementation strain (FC78). Strain FC57 was conjugated with plasmid pFC44 to insert a wild-type copy of mxaY into the ΔxoxFS strain prior to deleting the native copy of mxaY, and creating the final strain FC93 (ΔxoxFS ΔmxaY; METBUDRAFT_2794::PmxaY-mxaY).

Plasmids were constructed by Gibson assembly (Gibson et al., 2009). The plasmid pFC45 containing the mxaY E147G construct was constructed using Gibson assembly with a PCR product containing the mxaY E147G gene. The mxaY E147G PCR product was made by fusion of PCR products with primers containing the E147G mutation (A440G nucleotide mutation) (Shevchuk et al., 2004). All strains and plasmids are listed in Table 1. All primers are listed in Table S1.

RNA isolation and real-time qRT-PCR assays

RNA was isolated and checked for purity as previously described (Chu & Lidstrom, 2016). Briefly, cells were grown in the presence or absence of lanthanum and RNA was harvested using an acid phenol-chloroform extraction. We used iScript™ Reverse Transcription Supermix (Biorad) with and without reverse transcriptase to ensure all RNA was DNA-free. RNA quality and concentration was determined using Nanodrop® ND-1000. cDNA was generated using 100 ng–500 ng isolated RNA as template with the SensiFAST™ cDNA Synthesis Kit (Bioline). PCR reactions contained the following components: 400 µM primers, SensiFAST™ SYBR® No-ROX Kit (Bioline), cDNA, and ddH2O up to 10 µl volume. The PCR reactions were performed in LightCycler® Capillaries (Roche Diagnostics) and reactions were run using a LightCycler 2.0 (Roche Diagnostics). The following PCR program was used: Step 1: 95°C for 3 min, Step 2: 95°C for 10 s, Step 3: 55°C for 15 s, Step 4: 72°C for 45 s, Step 5: 72°C for 3 s (SYBR detection), Step 6: repeat steps 2 through 5 40 times. Ct values were determined using LightCycler Software Version 3.5 (Roche) and all gene expression values were normalized to 16S Ct values. All primers used for real-time quantitative reverse transcription (qRT)-PCR assays are listed in Table S1.

Results

ΔxoxFS mutant contains mutation in gene encoding MxaY

Previously, we identified a suppressor mutant of the M. buryatense 5GB1C ΔxoxF strain, named ΔxoxFS (Chu & Lidstrom, 2016). This suppressor mutant displayed a wild-type growth rate when cultivated in the presence of lanthanum, conditions under which the MxaFI MDH is normally not produced (Fig. 1). We identified the ΔxoxFS suppressor mutant allowed for the constitutive activation of the operon encoding MxaFI as well as its transcriptional regulator, MxaB. We submitted the ΔxoxFS mutant for whole genome sequencing (WGS) to attempt to identify the causative mutation of the broken lanthanide switch. In the ΔxoxFS mutant, we identified a mutation in the gene encoding a histidine kinase (locus tag METBUDRAFT_1818) that would convert a glutamate at amino acid positon 147 to a glycine. The mutated histidine kinase contains homology to genes encoding the MxaY histidine kinase. The histidine kinase in M. buryatense shares a 27% amino acid identity to P. denitrificans MxaY, 40% identity to Methylococcus capsulatus MxaY, and 51% identity to Methyloglobulus morosus annotated MxaY. The glutamic acid at amino acid position 147 is widely conserved, but lies outside of the domains involved in histidine kinase function (Galperin & Nikolskaya, 2007; Camacho et al., 2009). Interestingly in the P. denitrificans genome, the location of the mxaXYZ operon, upstream and divergently transcribed from the mxaF operon, is the same as the location of the mxaB gene in the M. buryatense genome.

Figure 1 Mutation in MxaY is responsible for ΔxoxFS suppressor phenotype.

Growth curves for wild-type M. buryatense 5GB1C and ΔxoxF(FC53), ΔxoxFS (ΔxoxF suppressor strain; FC57), and ΔxoxFS complemented with a wild-type copy of mxaY (ΔxoxFS; mxaY; FC93) grown in cultivation medium with (A, filled symbols) and without (B, empty symbols) 30 µM lanthanum. Data points represent the mean from three replicates and error bars represent standard deviation.

To determine if the E147G mutation in MxaY is the causative mutation in the ΔxoxFS strain, a wild-type copy of mxaY under the control of its native promoter was inserted into the ΔxoxFS chromosome at a location known to be transcriptionally silent (Yan et al., 2016). The existing copy of mxaY (containing the E147G mutation) was then deleted in this strain. The resulting strain (FC93) contains the ΔxoxF and ΔmxaY E147G deletions from their native locations and a heterologously expressed copy of wild-type mxaY. We determined the growth rate of the resulting mxaY complemented strain, compared to ΔxoxFS and the original ΔxoxF strain. The restoration of one wild-type copy of mxaY to the original ΔxoxFS strain resulted in a severe growth rate defect when the strain was grown with lanthanum (Table 2 and Fig. 1). This phenotype resembled that of the original ΔxoxF strain (Table 2 and Fig. 1). In contrast, the ΔxoxFS strain demonstrated a wild-type growth rate regardless of the presence or absence of lanthanum (Table 2 and Fig. 1) (Chu & Lidstrom, 2016). These results demonstrate that the MxaY E147G mutation was responsible for the suppressor phenotype observed in the ΔxoxFS strain.

Table 2 Doubling times of xoxF variant strains.a

Strain	With La3+	Without La3+	
WT 5GB1C	2.32 ± 0.17	2.75 ± 0.17	
ΔxoxF	0 ± 0	2.63 ± 0.15	
ΔxoxFS	2.24 ± 0.19	2.52 ± 0.14	
ΔxoxFS ΔmxaY E147G; PmxaY-mxaY	0 ± 0	2.58 ± 0.33	
Notes.

a Doubling times are in hours and represent the means of three technical replicates ± standard deviation. Doubling times were calculated from three time points during the exponential phase of growth, when possible.

TITLE WT wild-type

La3+ lanthanum

MxaY controls the lanthanide-mediated MDH switch

The mxaY gene encodes a histidine kinase whose homolog has been previously shown to be involved in mxaF regulation in P. denitrificans (Harms et al., 1993; Yang et al., 1995). In order to determine whether MxaY played a role in the lanthanide-mediated MDH switch in M. buryatense 5GB1C, an mxaY deletion mutation was generated. Real-time qRT-PCR was performed on RNA isolated from the ΔmxaY strain. Compared to wild-type M. buryatense5GB1C grown in the absence of lanthanum (when mxa operon transcription is normally active), the ΔmxaY strain grown in the absence of lanthanum shows a decrease in mxaF gene transcription and an increase in xoxF transcripts (Fig. 2A). The ΔmxaY phenotype is similar to that of the ΔmxaB mutant (Chu & Lidstrom, 2016), and indicates that MxaY also functions to upregulate transcripts for the MxaFI-type MDH and repress transcripts for the XoxF-type MDH. We also observed a decrease in the level of transcripts from the lanthanide-responsive mxaB in the ΔmxaY mutant compared to wild-type M. buryatense 5GB1C, indicating that MxaY also induces transcription of the orphan response regulator. The results from the ΔmxaY mutant strain suggest that MxaY is normally active when cells are cultivated in the absence of lanthanum.

Figure 2 MxaY regulates the lanthanide-mediated MDH switch.

(A) Real-time qRT-PCR was performed on RNA harvested from wild-type and ΔmxaY(FC74) M. buryatense 5GB1C strains grown in the absence of lanthanum. The values shown represent the fold change in mxaF, mxaB, and xoxF gene expression when transcript levels from the ΔmxaY strain were compared to wild-type M. buryatense 5GB1C. (B) Real-time qRT-PCR was performed on RNA harvested from the ΔmxaY mutant grown with and without lanthanum. Results shown represent the fold change in gene expression in ΔmxaY cells grown with lanthanum compared to gene expression in ΔmxaY cells grown without lanthanum. All gene expression was normalized to 16S rRNA transcript levels. Significant differences between gene expression levels in (A) and (B) were determined by using unpaired t-tests (∗∗∗p < 0.001; ∗∗p < 0.01; ∗p < 0.05). La3+, lanthanum. (C, D) Growth curves for wild-type M. buryatense 5GB1C, ΔmxaY (FC74), ΔmxaY complemented with a wild-type version of mxaY (ΔmxaY; mxaY; FC77), and ΔmxaY complemented with the E147G version of mxaY (ΔmxaY; mxaY E147G; FC78) strain variants grown in the presence (C, filled symbols) or absence (D, empty symbols) of 30 µM lanthanum. Data represent the means from three replicates and error bars represent standard deviations.

When transcripts from ΔmxaY mutant cells grown with lanthanum are compared to transcripts from ΔmxaY cells grown in the absence of lanthanum, the lanthanide-mediated mxa and xoxF transcriptional regulation disappears (Fig. 2B). Interestingly, xoxF transcription is not upregulated in a lanthanide-dependent manner in the ΔmxaY mutant. This is in contrast to the ΔmxaB mutant, in which we still observed a 3X increase in xoxF expression in the presence compared to the absence of lanthanum (Chu & Lidstrom, 2016). These results altogether suggest that MxaY functions upstream of MxaB in the regulatory pathway that controls the lanthanide-mediated MDH switch in M. buryatense. At this time, it is unclear whether MxaY phosphorylates MxaB directly or serves as part of a more complex regulatory cascade.

The ΔmxaY strain displayed impaired growth in the presence or absence of lanthanum (Table 3, Figs. 2C and 2D). When cultured in the absence of lanthanum, the ΔmxaY mutant has downregulated mxa transcripts and there is no lanthanum to activate XoxF MDH function. Interestingly, some ΔmxaY cultures still grew to high cell densities in the absence of lanthanum, although there was great disparity in the growth rate of the replicates (Table 3 and Fig. 2D). In the presence of lanthanum the ΔmxaY mutant still exhibits a severe growth defect (Table 3 and Fig. 2C), which suggests that MxaY has a pleiotropic effect on lanthanide-controlled metabolism. The ΔmxaY phenotype can be complemented by the addition of a chromosomally expressed copy of the wild-type mxaY gene under the control of its native promoter (Fig. S1, Table 3).

Table 3 Doubling times of mxaY variant strains.a

Strain	With La3+	Without La3+	
WT 5GB1C	2.44 ± 0.11	2.66 ± 0.35	
ΔmxaY	7.63 ± 1.87	4.78 ± 2.46	
ΔmxaY; PmxaY-mxaY	2.38 ± 0.16	2.80 ± 0.01	
ΔmxaY; PmxaY-mxaY E147G	7.47 ± 1.37	2.91 ± 0.07	
Notes.

a Doubling times are in hours and represent the means of three technical replicates ± standard deviation. Doubling times were calculated from three time points during the exponential phase of growth, when possible.

TITLE WT wild-type

La3+ lanthanum

Located downstream of mxaY is the gene, METBUDRAFT_1817, which is predicted to encode a response regulator. The protein product of gene METBUDRAFT_1817 is 50% identical to M. buryatense MxaB at the amino acid level. However, METBUDRAFT_1817 does not have significant homology to the response regulator MxaX, which in P. denitrificans is encoded by the gene downstream of mxaY and is required for mxaF gene transcription in Harms et al. (1993); Yang et al. (1995). Because of its location in the M. buryatense genome and its similarity to MxaB, we tested whether METBUDRAFT_1817 was involved in the lanthanide-mediated MDH switch. A deletion mutation in this gene did not alter the lanthanide-mediated regulation of mxa genes or xoxF compared to wild-type M. buryatense 5GB1C (Fig. S2) (Chu & Lidstrom, 2016).

MxaY E147G mutant is constitutively active

To determine the consequence of the causative mxaY suppressor mutation identified in the ΔxoxFS strain, a strain was constructed that contained only the E147G mutant version of MxaY. This strain was generated from the ΔmxaY mutant by integrating the gene encoding MxaY E147G at a heterologous site in the chromosome, under the control of the mxaY promoter. Real-time qRT-PCR was performed on RNA isolated from the MxaY E147G strain grown in the presence or absence of lanthanum. Compared to wild-type M. buryatense 5GB1C grown in the presence of lanthanum, conditions under which we predict MxaY is normally inactive, the MxaY E147G strain grown with lanthanum displays increased expression of mxaF and mxaB and decreased xoxF expression (Fig. 3A). The MxaY E147G strain exhibited similar levels of mxaF, mxaB, and xoxF expression regardless of the presence or absence of lanthanum (Fig. 3B). Together, these results suggest that the E147G mutation renders MxaY constitutively active.

Figure 3 MxaY E147G is a constitutively active version of MxaY.

(A) Real-time qRT-PCR was performed on RNA harvested from wild-type and MxaY E147G (FC78) M. buryatense 5GB1C strains grown in the presence of 30 µM lanthanum. The values shown represent the fold change in gene expression in MxaY E147G compared to wild-type M. buryatense 5GB1C. (B) Real-time qRT-PCR was performed on RNA harvested from MxaY E147G (FC78) cells grown in the presence or absence 30 µM lanthanum. Results represent the fold change in gene expression in cells grown with lanthanum compared to cells grown without lanthanum. All gene expression was normalized to 16S rRNA transcript levels. Significant differences between gene expression levels was determined by using unpaired t-tests (∗∗p < 0.01; ∗p < 0.05). Data represent the means from three replicates and error bars represent standard deviations. La3+, lanthanum.

In agreement with the above results, the MxaY E147G mutant has a wild-type growth rate when cultivated in the absence of lanthanum (Fig. 2D and Table 3), conditions under which we predict mxaY is normally activated. The MxaY E147G mutant surprisingly displayed a marked growth defect when grown in the presence of lanthanum (Fig. 2C and Table 3). This indicates that inappropriate inactivation of MxaY is detrimental to the cell.

Discussion

The transcription of genes encoding the MxaFI and XoxF MDH systems is tightly and differentially controlled by a lanthanide-mediated switch (Farhan Ul Haque et al., 2015b; Chu & Lidstrom, 2016; Gu et al., 2016; Vu et al., 2016). In this paper, we describe another regulator of the lanthanide-mediated switch, MxaY, that transcriptionally upregulates genes encoding the MxaFI MDH while downregulating xoxF in M. buryatense. MxaY is the most upstream factor in the lanthanide-regulatory cascade identified to date, as it also controls the transcription of the MxaB response regulator (Figs. 2A and 2B). It is not yet known whether MxaY and MxaB interact directly, or whether they are part of a more complex signaling cascade. Our findings are in contrast to the MDH regulatory cascade in P. denitrificans, in which a deletion of mxaY still allowed for normal mxaF transcription, perhaps because of a redundant histidine kinase (Yang et al., 1995). However, MxaY in P. denitrificans does play a role in activating mxaF expression as overexpression of MxaY resulted in increased mxaF transcription (Yang et al., 1995). In M. buryatense, MxaY seems to be the predominant histidine kinase regulating the lanthanide-mediated MDH switch, as the lanthanum response of mxaF and xoxF transcription was absent in the ΔmxaY mutant (Fig. 2B).

The growth rate of the ΔmxaY mutant is impaired when cultivated in both the presence and absence of lanthanum (Figs. 2C and 2D; Table 3). Previous studies have shown that MDH mutants grow at wild-type growth rates under cultivation conditions in which the alternative MDH is transcribed and active (Farhan Ul Haque et al., 2015a; Chu & Lidstrom, 2016; Vu et al., 2016). These growth rate results suggest MxaY function affects more than just MDH gene expression. In addition, the extent of the growth rate impairment is lanthanide-mediated, indicating that MxaY plays a larger role in the lanthanum response than just mxaFI and xoxF transcription. The identification of the MxaY regulon may aid in the discovery of other lanthanide-responsive genes, which may or may not be involved in MDH function and regulation.

The growth rate of ΔmxaY cultivated in the absence of lanthanum was higher than expected (Fig. 2D and Table 3), as mxaFI genes are downregulated and no supplemented lanthanum is present for the XoxF active site under these conditions. Several possible explanations exist for this observation. First, M. buryatense 5GB1C may be able to scavenge minute amounts of lanthanides present in the media or glassware to support limited growth. Second, alternative metals, such as calcium may be able to poorly substitute for lanthanides in the XoxF active site. Last, the high standard deviation of the ΔmxaY mutant growth rate in the absence of lanthanum could point to the presence of spontaneous suppressors arising. It is possible these suppressor mutations may allow XoxF to better utilize alternative metals in its active site.

Our results suggest the identified MxaY E147G mutation allows for the constitutive activation of MxaY, regardless of the presence or absence of lanthanum (Fig. 3). As this amino acid residue lies outside of the domains required for histidine kinase function but is highly conserved amongst methylotrophic homologues, it is tempting to speculate this E147 residue is involved in the lanthanide response. The suppressor mutants of the M. buryatense 5GB1C ΔxoxF mutant arose easily (Chu & Lidstrom, 2016), and perhaps this means multiple mutations in MxaY can result in constitutive activation. If MxaY is directly sensing lanthanum, E147 and other potential residues identified from ΔxoxFS variants may lead to the identification of a lanthanide-binding motif.

The constitutive activation of MxaY in the MxaY E147G strain resulted in a growth rate defect when the strain was cultivated in the presence of lanthanum (Fig. 2C and Table 3). Curiously, the ΔxoxFS strain also contains a constitutively activated MxaY copy, but has a wild-type growth rate in the presence of lanthanum (Fig. 1A and Table 2) (Chu & Lidstrom, 2016). In strains with MxaY constitutively activated, genes encoding MxaFI will be transcribed and the enzyme active because of the presence of calcium in the cultivation medium. The growth rate results suggest the additional presence of an active XoxF MDH, which is present in the MxaY E147G complementation strain but not in ΔxoxFS, is detrimental to the cell. Despite the fact that MxaY represses XoxF, some active XoxF is predicted to be present in the MxaY E147G strain, since we previously have shown that in an ΔmxaF mutant the repressed level of XoxF in the absence of supplemental lanthanum allows for some growth (Chu & Lidstrom, 2016). Our results suggest that this level of active XoxF, combined with active MxaFI result in a drastic reduction in growth rate (Fig. 2C and Table 3). This is the first evidence suggesting that tight control of the MDH enzymes is required for proper metabolic function. Uncontrolled activation of both MDH enzymes might be detrimental to growth if the high levels of transcription and translation of the MDH components places too great a burden on cells.

Conclusion

The use of methanotrophs as industrial microorganisms is an attractive method of producing value-added chemicals while utilizing excess atmospheric methane. The economic feasibility of this approach depends upon successful metabolic engineering strategies, which in turn depends on an in-depth knowledge of methanotrophic metabolism and its regulation. The expression of the two forms of MDH is an example of regulation that could have an impact on metabolic engineering. Here we identified an additional regulator, the histidine kinase MxaY, of the lanthanide-mediated MDH switch in M. buryatense. We identified a SNP located outside of the conserved histidine kinase region that allows for constitutive activation of MxaY. This information could potentially lead to identification of a lanthanide response or binding domain. Currently, there is great interest in studying the role of lanthanides in methylotrophic metabolism and identification of a lanthanide sensor and binding domain may result in the discovery of additional lanthanide-mediated functions in methylotrophic microorganisms.

Supplemental Information

Figure S1 Complementation of ΔmxaY restores lanthanide-mediated MDH gene expression

Real-time qRT-PCR was performed on RNA harvested from the ΔmxaY wild-type complementation strain (FC77) grown in the presence or absence of 30 µM lanthanum. Results shown represent the fold change in gene expression in cells grown in the presence of lanthanum compared to gene expression in cells grown in the absence of lanthanum. Gene expression was normalized to 16S rRNA transcript levels. Unpaired t-tests were used to determine significance in gene expression levels between the two conditions tests (∗∗∗p < 0.001, ∗∗p < 0.01). Data represent the means from three replicates and error bars represent standard deviations. La3+, lanthanum.

Click here for additional data file.

Figure S2 Predicted response regulator METBUDRAFT_1817 is not required for lanthanide-mediated MDH switch

Real-time qRT-PCR was performed on RNA harvested from Δ MBURv2_1817 (FC70) M. buryatense 5GB1C cells cultivated in the presence or absence of 30 µM lanthanum. Results shown represent the fold change in gene expression in cells grown in the presence of lanthanum compared to gene expression in cells grown in the absence of lanthanum. Gene expression was normalized to 16S rRNA transcript levels. Unpaired t-tests were used to determine significance in gene expression levels between the two conditions tests (∗∗p < 0.01). Data represent the means from three replicates and error bars represent standard deviations. La3+, lanthanum.

Click here for additional data file.

Table S1 Primers used in this study

Click here for additional data file.

Data S1 MxaY raw data

Raw data for both growth curves and real-time qRT-PCR assays

Click here for additional data file.

We thank L Chistoserdova for helpful discussions and critical reading of the manuscript. We thank M Pesesky, A Gilman, Y Fu, and AW Puri for useful discussions.

Additional Information and Declarations

Competing Interests

Author Contributions

DNA Deposition

Data Availability

The authors declare there no competing interests.

Frances Chu conceived and designed the experiments, performed the experiments, analyzed the data, contributed reagents/materials/analysis tools, wrote the paper, prepared figures and/or tables, reviewed drafts of the paper.

David A.C. Beck conceived and designed the experiments, analyzed the data, contributed reagents/materials/analysis tools, reviewed drafts of the paper.

Mary E. Lidstrom conceived and designed the experiments, contributed reagents/materials/analysis tools, reviewed drafts of the paper.

The following information was supplied regarding the deposition of DNA sequences:

NCBI BioProject Accession number: PRJNA322692.

The following information was supplied regarding data availability:

The raw data has been supplied as a Supplemental Dataset.

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
