# Peer review of "MxaY regulates the lanthanide-mediated methanol dehydrogenase switch in Methylomicrobium buryatense"

_PeerJ, doi:10.7717/peerj.2435_

## Round 0.1 · original submission · Minor Revisions

As you will see, all three expert reviewers considered your manuscript an important and interesting contribution to the field, and provided some minor suggestions for further improvement.

·

Basic reporting

No comments

Experimental design

The experimental design is clear and sound and describes original research.

Validity of the findings

The data presented are robust and statistically sound.

Additional comments

The article is a follow up on Chu & Lidstrom 2016 (J. Bacteriol.) in which they showed that XoxF acts as the predominant methanol dehydrogenase in M.buryatense together with the involvement of MxaB in the regulation of the expression of this enzyme. In this study they also identified a surpressor mutant, delta-xoxFS, which is studied in detail in the current article. The results obtained clearly show the complexity of the regulation of both types of methanol dehydrogenases (mxaF/xoxF) by presence and absence of lanthanides. MxaY is identified as an additional regulator. The fast-growing microorganism combined with the available impressive genetic toolbox holds a big promise for future research.
I only have some minor textual remarks:
- Please replace ‘discover’ by another verb (lines21, 107, 162)
- Line 57: delete ‘generally’
- Line 68-71: in strain SolV, xoxF is the only methanol dehydrogenase.
- Line 161: insert reference Chu & Lidstrom, 2016

Reviewer 2 ·

Basic reporting

1. Citations need to be formatted properly. For example, "Fei, Guarnieri et al. 2014" should be formatted as "Fei et al., 2014" in the main text. The reference section needs to be reformatted as well.

2. Accession numbers for the genome sequence of M. buryatense need to be presented.

3. Distinction between lanthanum and lanthanides is unclear throughout the manuscript. The authors needs to clarify whether they meant 'lanthanum' as the specific element or 'lanthanide' as a group of elements.

4. Introduction is a little disorganized and unfocused.

Experimental design

1. All reports of qPCR or RT-qPCR experiments should conform to MIQE guidelines. A large portion of necessary information are missing in the manuscript.

2. More specifically, RT-qPCR requires an internal standard. Transcription level of a housekeeping gene with constitutive expression can be used for normalization of RT-qPCR data. Another method is to use control mRNA to measure the recovery rate of RNA, which allows absolute quantification of the target mRNA.

Validity of the findings

No comment

Additional comments

1. The figure legends are very confusing. Readers will have hard time understanding what "(delta)xoxFS; mxaY" means in Figure 1A and B without referring back to the manuscript. This comment also applies to Figure 2C and D.

·

Basic reporting

The quality of language is largely excellent. Figures and tables are designed with care.

The whole study provides an important piece in mosaic of the regulatory network needed to control PQQ-dependent alternative MDH synthesis/expression, i.e. XoxF, in methanotrophs. It convincingly describes the newly revelaed role of the regulatory protein MxaY.

The study provides an important step forward in understanding the lanthanoid switch in methylotrophs.

Experimental design

The studies has a clear scientific objective that has not been adressed previously. Associated methods are well conducted and described.

Validity of the findings

The major finding that mxaY encodes for a previously undiscovered regulatory element in XoXF synthesis is thoroughly deduced from convincing and complementary experiments.

It would have been nice to see also data of binding of lanthanum to purified MxaY, which would then clearly demonstrate that MxaY senses lanthanum, a topic on which the authors speculate but could not resolve with their experimental approach.

Additional comments

Please;...
... avoid saying '...as fig XY shows...'. Just mention the fact/observation and state the figures/tables in pararentheses. Please, change that in the whole manuscript.
... it is not evident for the reviewer how a methanotroph that has not the capability to utilize atmospheric methane can contribute (when used in industrial production) to mitigate the atmospheric burden of methane (lns 308-309). Please rephrase or better explain? Is that argumentation needed for the story?

---

## Round 0.2 · accepted · Accept

I have been asked to take over editorial duties by the Publisher as the original Editor is unavailable.

This is a well written and interesting manuscript and the reviewers have indicated that there are no further points that need to be addressed.

A note on the references. The PeerJ guidelines state:

"We include reference formatting as a guide to make it easier for editors, reviewers, and preprint readers, but will not strictly enforce the specific formatting rules as long as the full citation is clear. Styles will be normalized by us if your manuscript is accepted"

The reference in the manuscript are in a consistent format and therefore do not need to be addressed further by the authors, (although in future submissions the authors should try and adhere to in-house style).

·

Basic reporting

No additional comments

Experimental design

No additional comments

Validity of the findings

No additional comments

Additional comments

I fully satisfied with the changes made in the manuscript, addressing all remarks of the reviewers.

Reviewer 2 ·

Basic reporting

Most of the reviewers' comments were addressed and the manuscript generally looks to be in a much better form.

Experimental design

The authors added description in the methods section as an effort to conform to the MIQE guidelines but some information is still missing from either the main text or in supplementary information. For example, calibration curve data and the lower limit of detection need to be mentioned in the manuscript according to the guidelines.

Validity of the findings

In the revised form, the findings of the experiments are clearer and easier to comprehend.

Additional comments

The authors dismissed reviewer's comments regarding the reference format. For references with more than two authors, 'Author A, Author B et al. 20XX)' is in an obviously incorrect format and I see the same misformatting throughout the manuscript.

PeerJ author guidelines clearly state that the correct reference format is as shown in the example below.
"Example journal reference: Smith JL, Jones P, Wang X. 2004. Investigating ecological destruction in the Amazon. Journal of the Amazon Rainforest 112:368-374. DOI: 10.1234/amazon.15886."
This format is clearly different from what I see in the manuscript. Please pay closer attention to the author guidelines.